# Evaluation of Antibacterial Activity against Nosocomial Pathogens of an Enzymatically Derived α-Aminophosphonates Possessing Coumarin Scaffold

**DOI:** 10.3390/ijms241914886

**Published:** 2023-10-04

**Authors:** Paweł Kowalczyk, Dominik Koszelewski, Anna Brodzka, Karol Kramkowski, Ryszard Ostaszewski

**Affiliations:** 1Department of Animal Nutrition, The Kielanowski Institute of Animal Physiology and Nutrition, Polish Academy of Sciences, Instytucka 3, 05-110 Jabłonna, Poland; 2Institute of Organic Chemistry, Polish Academy of Sciences, Kasprzaka 44/52, 01-224 Warsaw, Poland; anna.brodzka@icho.edu.pl (A.B.); ryszard.ostaszewski@icho.edu.pl (R.O.); 3Department of Physical Chemistry, Medical University of Bialystok, Kilińskiego 1 Str., 15-089 Białystok, Poland; kkramk@wp.pl

**Keywords:** α-aminophosphonates, coumarins, Kabachnik–Fields reaction, Gram-positive bacteria, Gram-negative bacteria, lipases, antimicrobial activity, nosocomial infections, MIC

## Abstract

The purpose of the present study was to evaluate the synergistic effect of two important pharmacophores, coumarin and α-amino dimethyl phosphonate moieties, on antimicrobial activity against selected strains of multidrug-resistant nosocomial pathogenic bacteria. The previously developed enzyme-catalysed Kabachnik–Fields protocol allowed us to obtain the studied compounds with high yields which were free from metal impurities. The structure–activity relationship revealed that inhibitory activity is strongly related to the presence of the trifluoromethyl group (CF_3_−) in the coumarin scaffold. MIC and MBC studies carried out on six selected pathogenic bacterial strains (Gram-positive pathogenic *Staphylococcus aureus* (ATCC 23235) strain, as well as on Gram-negative *Acinetobacter baumannii* (ATCC 17978), *Pseudomonas aeruginosa* (ATCC 15442), *Enterobacter cloacae* (ATCC 49141), *Porphyromonas gingivalis* (ATCC 33277), and *Treponema denticola* (ATCC 35405)) have shown that tested compounds show a strong bactericidal effect at low concentrations. Among all agents investigated, five exhibit higher antimicrobial activity than those observed for commonly used antibiotics. It should be noted that all the compounds tested showed very high activity against *S. aureus*, which is the main source of nosocomial infections that cause numerous fatalities. Furthermore, we have shown that the studied coumarin-based α-aminophosphonates, depending on their structural characteristics, are non-selective and act efficiently against various Gram-positive and Gram-negative pathogens, which is of great importance for hospitalised patients.

## 1. Introduction

Nosocomial infections are one of the most common complications in immunosuppressed cancer patients. Today, the extensive spread of multidrug-resistant bacteria (MDR) among humans, animals, and environmental reservoirs has created new unanticipated epidemiological patterns [1]. Hospitalised patients are also affected by MDR infections, often associated with considerable morbidity, mortality, and financial burden [2]. Meticillin-resistant *Staphylococcus aureus* (*S. aureus*) (MRSA), vancomycin-resistant enterococci, and multidrug-resistant Gram-negative bacteria, such as pathogenic *Enterobacter* spp., are common pathogens. In all countries, some routine surgical operations and cancer chemotherapy will become less safe without effective antibiotics to protect against infections. Incorrectly selected antibiotics will not only be ineffective, but may also have a number of side effects, while the infection continues to develop, weakening the patient’s body more and more.

Despite efforts, outbreaks continue to occur. The urgent need for action is consistent with a precautionary approach, and national and international multisectoral action and collaboration should not be impeded by gaps in knowledge [3]. Nosocomial pathogens under favourable conditions can cause dangerous local or systemic infections. *Staphylococcus* infections can affect many areas of the human body. These bacteria can also attack subcutaneous tissues and soft tissues and even contribute to systemic infections such as endocarditis and inflamed heart valves [4,5,6,7,8]. Therefore, it is necessary to develop effective antimicrobial drugs [9]. As a result of the progressive increase in the resistance of *S. aureus* to methicillin (MRSA), vancomycin is the most widely used drug in infections. The emergence of vancomycin-resistant *S. aureus* infections means that the treatment of staph infections can become much more difficult in the future. Unlike Gram-negative bacteria, the Gram-positive pathogens develop a thick layer of peptidoglycan (PG) that surrounds the plasma membrane and protects bacteria from an unpredictable and often hostile environment (Figure 1). Furthermore, the PG layers of many Gram-positive bacteria are densely functionalised with anionic glycopolymers known as wall teichoic acids (WTAs), making them even more resistant to potential antimicrobial agents [10].

Recently, pharmacologists have focused their attention on new therapeutic agents such as coumarins [11,12,13]. Furthermore, compounds containing the structural motif 2*H*-chromen-2-one are well known in a variety of natural products [14,15,16,17] (Figure 1).

Based on the literature data, naturally occurring coumarins have been identified as privileged groups of compounds with antimicrobial activity (Figure 1). Therefore, it can be assumed that compounds containing the coumarin structural element in their structure can potentially be used as antimicrobial agents [18,19,20,21,22,23,24]. It is well recognised that the substitution of a hydrogen atom with fluorine in a molecule introduces minimal steric alterations due to its relatively small size, a fact that can facilitate interactions of a fluorinated biomolecule with the receptor site [25,26,27,28]. In addition, the introduction of fluorine as a most electronegative atom, which modulates the electron density of coumarin and participates as a hydrogen acceptor enhancing lipophilicities, can usually significantly alter the physicochemical properties of the molecule (such as solubility or log (*P*)). This could improve an antibacterial activity and stability. It can also reduce toxicity to eukaryotic cells, which improves therapeutic efficiency. In addition, substitution of hydrogen by a trifluoromethyl group could act in a similar way. In this case, the size of the CF_3_- group is much larger, making its contribution to hydrophobicity even greater [29,30]. The limited availability of antibiotics to treat multidrug-resistant Gram-negative and Gram-positive bacterial infections remains a serious problem. Therefore, it is imperative to develop new agents or new therapeutic strategies that are able to overcome drug resistance in these organisms. To reach this goal, there is an urgent need to design agents that are able to penetrate both outer and inner membranes protecting mentioned pathogens. One approach to overcome this involves appending or tweaking different functional groups on a lead structure, in the hope of generating more amenable derivatives with enhanced biological activity and cellular permeation. It was recently shown that, to accumulate in the Gram-negative bacteria, *E. coli* small molecules must contain an amine group, be amphiphilic, be rigid, and have low globularity [31]. α-Aminophosphonates are among the most common and biologically active organophosphorus compounds [32]; those with heterocyclic moieties especially show very interesting biological activities and have been used as antibacterial agents [33,34,35,36,37,38]. The appropriate combination of coumarin and phosphonate scaffolds was found to lead to the formation of compounds with the desired synergistic antimicrobial properties [39,40,41]. Our previous studies revealed that compounds possessing an aminocoumarin scaffold with fluorine and trifluoromethyl groups were effective inhibitors of the growth of various strains of LPS *E. coli* [42,43,44]. Thus, the objective of the present work is to identify the characteristics of the coumarin α-amino phosphonate structure that could be important for antibacterial activity against selected pathogenic strains with a particular emphasis on Gram-positive *S. aureus* and Gram-negative *Enterobacter* spp., which are responsible for nosocomial infection in all over the world. Moreover, we wanted to find compounds with both high antimicrobial activity and low selectivity for various types of pathogens. This is particularly important in cases of patients hospitalised due to infections with multidrug-resistant pathogens.

## 2. Results and Discussion

### 2.1. Chemistry

The most widely used and recognised method for the synthesis of α-aminophosphonates is the multi-component Kabachnik–Fields reaction. The usual synthetic protocol includes the condensation of equimolar amounts of three components (aldehyde, amine, and dialkyl phosphate) in organic solvents in the presence of various catalysts, mostly Lewis and Brönsted acids [45]. Unfortunately, many of the reported protocols suffer from some inconveniences, such as the stoichiometric amounts of catalysts (e.g., cancerogenic amines) or the application of toxic metal catalysts. Due to special requirements for the synthesis of biological compounds free from metal contamination, we focus on enzymatic methods [46]. Enzymes are very effective catalysts and have the advantage of being heavy-metal-free [47,48]. The target coumarin-based α-aminophosphonates **1**–**13** were prepared with yields of up to 92% according to the published procedure [42] (Figure 2). The structures of the compounds obtained were confirmed by NMR, provided in the Appendix A, and were consistent with published data.

### 2.2. Cytotoxic Studies of the Library of Coumarin α-Aminophoshonates ***1***–***13*** and Coumarin ***14***

The most common and serious MDR pathogens have been encompassed within the acronym “ESKAPE”, standing for *Enterococcus faecium* (*E. faecium*), *S. aureus*, *Klebsiella pneumoniae* (*K. pneumoniae*), *Acinetobacter baumannii* (*A. baumannii*), *Pseudomonas aeruginosa* (*P. aeruginosa*), and *Enterobacter* spp. (*E.* spp.) [49]. The antimicrobial agents available on the market have various drawbacks, such as toxicity, a narrow spectrum of activity, and the fact that some also exhibit drug–drug interactions. In view of the high incidence of infections in immune-affected patients, the demands for new antimicrobial agents with a broad spectrum of activity and good pharmacokinetic properties have increased. Thus, to achieve this goal, we decided to test coumarin α-aminophoshonates **1**–**14** as potential antimicrobial agents on the selected bacterial strains. The analysis of the MIC and MBC assays shows that, for all compounds tested, colour changes were observed on 48-cell culture plates [50]. The strains of Gram-positive *S. aureus* (ATCC 23235) are the most susceptible to a modification with these compounds (visible dilutions of 10^−3^ corresponding to a concentration of 0.52 µM).

The noxious activity of the 14 agents examined (Figure 2) is determined in bacterial cells, according to a previously published protocol [42], after the analysis of the MIC and MBC tests, for which the MIC values were recorded in the range of 2–8.3 µM, and the MBC values in the range of 2–9 (+/−0.5) µM for the Gram-negative investigated strains of *A. baumannii* (ATCC 17978), *P. aeruginosa* (ATCC 15442), *Enterobacter cloacae* (*E. cloacae*) (ATCC 49141), *Porphyromonas gingivalis* (*P. gingivalis*) (ATCC 33277), and *Treponema denticola* (*T. denticola*) (ATCC 35405) (Figure 3, Figure 4 and Figure 5). The minimal inhibitory concentration (MIC) defines in vitro levels of susceptibility or resistance of specific bacterial strains to the antibiotic applied. MBC is the minimal antibacterial density necessary to kill bacteria, that is, bactericidal, rather than purely bacteriostatic densities [43].

The results of our studies show that the synthesised compounds **1**–**13** can potentially be applied as ‘alternatives’ for antibiotics currently used in hospital and clinical infections. Therefore, studies on the antibacterial activity of newly synthesised compounds are of great importance in nosocomial or clinical infections [42,44]. Comparing the MIC values for the tested compounds **1**–**12** versus compound **13**, we see a clear impact of the CF_3_ group present in the coumarin structure on the increasing antimicrobial activity (Figure 4). Additionally, the introduction of a phosphonate moiety into the studied structures significantly increases the activity against all pathogens tested (compounds **1**–**13** vs. **14**). As we can see in Figure 3, the changes in the base structure of coumarin **14** are manifested by slight changes in the activity of the investigated α-aminophosphonates **1**–**14**, which may indicate that the coumarin subunit is mainly responsible for the activity of these compounds against Gram-positive *S. aureus* (ATCC 23235). However, in the case of Gram-negative pathogens, we observe a significant dependence of antimicrobial activity on the nature of the substituent located in the phenyl ring of the revised compounds **1**–**13**. The introduction of electron-donating groups as well as a halogen atom such as bromine or fluorine into the para-position of the phenyl ring of the studied compounds **1**–**5** increases the antimicrobial activity against selected Gram-negative bacteria. Note that, for other halogens, iodine, and chlorine, and electron-donating groups at the *para*-position, we observe a reduction in antimicrobial activity. We observe the same effect for heteroaromatic substituents and naphthyl group **6**–**12**. We also see the influence of the location of the electron-withdrawing substituent in the aromatic ring on the antimicrobial activity. The compound with a nitro group in the *meta*-position shows higher activity against *A*. *baumannii* (ATCC 17978) and *P*. *aeruginosa* (ATCC 15442) than its analogue, with an identical group in the *para*-position (compounds **8** and **9**). Finally, to our delight, tested compounds **4** and **5** reveal desired nonselective antimicrobial activity against all tested pathogens. This feature is particularly important in the fight against multidrug-resistant pathogens belonging to both Gram-positive and Gram-negative bacteria.

It should also be noted that in most cases the tested α-aminophosphonates showed higher antimicrobial activity than those of widely used antibiotics (Figure 3 and Figure 4). This is particularly important because there is a clear increase in the resistance of tested pathogens to antibiotics such as ciprofloxacin (cipro) or cloxacillin (clox) (Figure 4). It is not known in what time interval pathogens will develop resistance to bleomycins (bleo), which will significantly disrupt the arsenal of available antibiotics used in the fight against hospital infections.

Usually, antimicrobial agents are classified as bactericidal or bacteriostatic. If the MBC-MIC ratio is small (less than 4–6), a drug is considered bactericidal, and it is possible to obtain drug concentrations that kill 99.9% of the organisms exposed. If the ratio of MBC to MIC is large, it may not be possible to administer doses of the drug safely to kill 99.9% of the bacteria, and the drug is considered bacteriostatic. For many compounds, the distinction between bactericidal and bacteriostatic is not exact and depends on the drug concentration attained in the target tissue and the pathogen involved. In the case of all tested compounds **1**–**14**, we are dealing with bactericidal agents (Figure 6).

### 2.3. Analysis of Bacterial DNA Isolated from S. aureus (ATCC 23235) Strain Modified with Tested Coumarin α-Aminophoshonates

The results of our current studies (MIC values) and those previously obtained [42] revealed that coumarin α-aminophoshonates show a strong toxic effect on the analysed strains of model pathogens. The compounds numbered **1**–**5** were revealed to be the most active against studied pathogenic cells, similar to those observed for commonly used antibiotics. (Figure 7). As can be seen in Figure 3, the studied strain of *S. aureus* (ATCC 23235) was sensitive to all compounds tested. The antimicrobial activity was similar to or even higher than those observed for bleomycin used as a reference (Figure 7).

Modified bacterial DNA was digested with Fpg as previously described [42,44]. All selected coumarin derivatives **1**–**14** (Figure 7) can strongly change the topology of bacterial DNA. After a digestion with Fpg, the base excision repair (BER) system, approximately 3.5% of the oxidative damage was identified, which indicates very strong oxidative damage in bacterial DNA. The results obtained for the individual compounds were statistically significant at the level of *p* < 0.05 (Table 1).

To study an oxidative stress, modified bacterial DNA was digested with a marker of oxidative stress Fpg protein from the group of repair glycosylases [42,44]. We investigated whether the resulting modifications in bacterial DNA would introduce an oxidative damage to the three DNA chain by changing the topological forms of bacterial DNA: ccc, oc, and linear forms. These results were in agreement with our previous studies [43,44]. Modification of bacterial DNA with the studied compounds led to the observation that all analysed coumarin α-aminophoshonates can strongly affect the topology of bacterial DNA, even after digestion with Fpg protein on the basis of agarose gels. The sensitivity of the *S. aureus* (ATCC 23235) strain to the cytotoxic effect of the compounds used and after digestion of the Fpg protein was similar to that of *E. coli* strains as follows: R4 > R2 > R3 > K12. This remains in agreement with our previous studies [43,44]. Blocking BER enzymes stops DNA replication and causes apoptosis of bacterial cells. In the future, cytotoxicity studies will also be performed using various cell lines and cultures to assess the biocompatibility of test compounds with active coumarin α-aminophoshonates. The dysfunction of two membranes in the model bacterial strains is an ideal model to assess the effectiveness of these compounds in relation to the antibiotics used [43,44,45,46,47,48,49,50,51,52,53,54,55,56,57,58,59], (Figure 8).

The mechanism of action of given chemical compounds including an α-aminophosphonates-possessing coumarin scaffold on the cells of the analysed bacteria occurs by inducing strong oxidative stress in them, consisting of the initial adsorption of the compound molecules on the cell surface of Gram-positive and Gram-negative bacteria, and then its gradual penetration and accumulation of the substance inside the microbial cell, to a level which is sufficient to obtain a toxic effect. Next comes the stage of the penetration of the compound, which depends on both the physicochemical properties and the structure of the cell covers, one of the basic functions of which is protection against the toxic activity of chemicals [51,52,53,54,55,56,57,58,59]. Therefore, the level of antimicrobial effectiveness of different classes of chemical compounds against particular groups of microorganisms varies. These compounds destabilise the bacterial cell membrane, making it more permeable to its own molecules. The compounds analysed in Gram-negative bacteria may disrupt the biosynthesis of the cell wall structure due to the binding of magnesium ions that stabilise lipopolysaccharide (LPS). The removal of Mg^2+^ ions by chelating compounds such as EDTA causes the release of membrane vesicles containing LPS from cells and changes the permeability of the outer membrane of bacteria [51,52,53,54,55,56,57,58,59]. In addition to disturbances in the permeability of the cytoplasmic membrane, the analysed compounds can cause the activation of autolytic enzymes and induce the formation of free radicals. The molecules of the compounds analysed enter the cytoplasm of bacterial cells, interacting with proteins and nucleic acids, causing disruption of the proper course of the replication process or cellular respiration leading to the denaturation of cytosolic proteins and nucleic acids in the cell [51,52,53,54,55,56,57,58,59].

## 3. Materials and Methods

### 3.1. Microorganisms and Media

The reference bacterial strain of *Staphylococcus aureus* (ATCC 23235) was provided from (LGC Standards UK) and was used according to the recommendation of ISO 11133: 2014. *Acinetobacter baumannii* (ATCC 17978), *Pseudomonas aeruginosa* (ATCC 15442), *Enterobacter cloacae* (ATCC 49141), *Porphiromonas gingivalis* (ATCC 33277), and *Treponema denticola* (ATCC 35405) strains were provided from the Medical University of Bialystok, Poland. These strains were used to test the antibacterial activity of the synthesised agents which is precisely described in [42]. These strains were used to test the antibacterial activity of the synthesized agents. Bacteria were cultivated in a tryptic soy broth (TSB; Sigma-Aldrich, Saint Louis, MI, USA) liquid medium and on agar plates containing TSB medium at 25 °C. Alternatively, TSB agar plates were used. The specific growth rate (μ) according to first-order kinetics was measured using a microplate reader (Thermo, Multiskan FC, Vantaa, Finland) at 605 nm in TSB medium. Lanes 1kb-ladder and Quick Extend DNA ladder (New England Biolabs, Ipswich, MA, USA) were used for the MIC and MBC tests, as described in detail in the previous work [42].

### 3.2. Minimum Inhibitory Concentration (MIC) and Minimum Bactericidal Concentration (MBC)

MIC and MBC were estimated using a microtiter plate method using sterile 48- or 96-well plates, which is precisely described in [42,43,44]. Briefly, MIC and MBC, defined as the lowest concentration of a bacteriostatic agent, were determined using a microtiter plate method using sterile 48- or 96-well plates. Briefly, 50 μL of the analysed compounds and the appropriate bacterial strains were added to the first row of the plate. Then, 25 μL of sterile Tryptone Soya Broth (TSB) medium was added to the other wells and serial dilutions were performed. Subsequently, 200 μL of inoculated TSB medium containing resazurin (0.02 mg/mL) as an indicator was added to all wells. TSB medium was inoculated with 10^6^ colony-forming units (CFU)/mL (approximately 0.5 McFarland units) of the bacterial strains. Plates were incubated at 30 °C for 24 h. The changes in colour from blue to pink or yellowish with turbidity were considered positive, and the lowest concentration at which no visible change in colour occurred was MIC according to Kowalczyk et al. [44]. Each experiment (both MIC and MBC) was repeated at least three times.

To estimate MBC, a dehydrogenase activity measurement was determined by measuring the visible changes in colour of triphenyl tetrazolium chloride (TTC) to triphenyl formazan (TF). Measures of 4 mM of dense culture (approximately 10^9^ CFU/mL) incubated in TSB medium at 25 °C for 24 h were placed in identical test tubes. Appropriate compounds were then added to the test tubes until the mixture reached a final concentration of 10–250 mg/mL. Then, the cultures were incubated at 30 °C for 1 h. The test tubes were then sealed with parafilm and incubated for 1 h at 30 °C in the dark. The lowest concentration at which no visible red colour (formazan) appeared was taken as MBC.

### 3.3. Statistical Analysis

All experimental data from at least three different trials (*n* = 3) were given as means standard error (SEM, manufacturer, Saint Louis, MO, USA). To compare pairs of means, the Tukey post hoc test was used, indicating statistical significance with * *p* < 0.05, ** *p* < 0.1, and *** *p* < 0.01 [44].

### 3.4. Chemicals

All reagents and solvents were purchased from Sigma-Aldrich, which is precisely described in [44].

### 3.5. General Procedure for the Synthesis of Coumarin α-Aminophosphonates ***1***–***13***

A mixture of the corresponding 7-amino-4-trifluoromethyl coumarin or 7-amino-4-methyl coumarin (1 mmol), the corresponding aldehyde (1 mmol), dimethyl phosphite (1 mmol), and Novozym 435 (80 mg) (2-Me-THF (2 mL) if noted; see Figure 2) was shaken at 200 rpm at 30 °C for 18 h. After the reaction was complete, the catalyst was separated on a glass frit funnel. The residue was washed with ethyl acetate. The combined organic phase was concentrated under vacuum. The resulting residue was purified by column chromatography (silica gel, eluent: ethyl acetate/hexanes, 6:4) to obtain the target coumarin α-aminophosphonates **1**–**13**. The yields of the derivatives are shown in Figure 2. The structures of the products were identified by their ^1^H NMR and were similar to those obtained previously [44].

## 4. Conclusions

The antimicrobial activity of coumarin α-aminophosphonates prepared by the enzyme-catalysed Kabachnik–Fields reaction against pathogenic bacterial strains responsible for nosocomial infections was investigated. Antimicrobial activity was enhanced by the synergetic effect of two pharmacophores, coumarin, and α-amino phosphonate. The enhanced effect on the antimicrobial activity of the CF_3_ group present in the coumarin scaffold against model strains of *S. aureus* (ATCC 23235), as well as on Gram-negative *A. baumannii* (ATCC 17978), *P. aeruginosa* (ATCC 15442), *E. cloacae* (ATCC 49141), *P. gingivalis* (ATCC 33277), and *T. denticola* (ATCC 35405) was revealed. Moreover, a strong impact of substituents located on the phenyl ring on antimicrobial activity was observed. The conducted studies show that the activity of tested compounds may result from changes in the spatial structure of bacterial cell membranes that cause their death. Among the coumarin α-aminophosphonates, compounds **1**–**5** showed super-selectivity and exhibited the highest cytotoxic activity, comparable to or better than antibiotics: ciprofloxacin, bleomycin, and cloxacillin. We have shown that the tested α-aminophosphonates are compounds exhibiting high antimicrobial activity with low selectivity to target pathogens, which plays an important role in the hospitalisation of patients and in the fight against multidrug-resistant bacteria. It should be noted that the low value of the MBC–MIC ratio indicates high toxicity of the compounds tested against the pathogenic *Staphylococcus aureus* ATCC 23235. Thus, the desired biocidal effect is obtained at very low concentrations of the tested compounds. The application of the investigated agents leads to strong oxidative stress and inhibition of the replication apparatus, which is extremely important in the case of pathogenic strains. The search for new nontoxic compounds for human health based on coumarin derivatives and toxic to bacterial cells will become an important factor in broadly understood antibacterial therapy in the emerging new pandemics of our century. In addition, breaking resistance to antibiotics by new environmentally modified bacterial strains created by the so-called super-resistant bacteria may be a breakthrough in the neutralisation and cytotoxicity of the analysed pathogenic species.

## Figures and Tables

**Figure 1 ijms-24-14886-f001:**
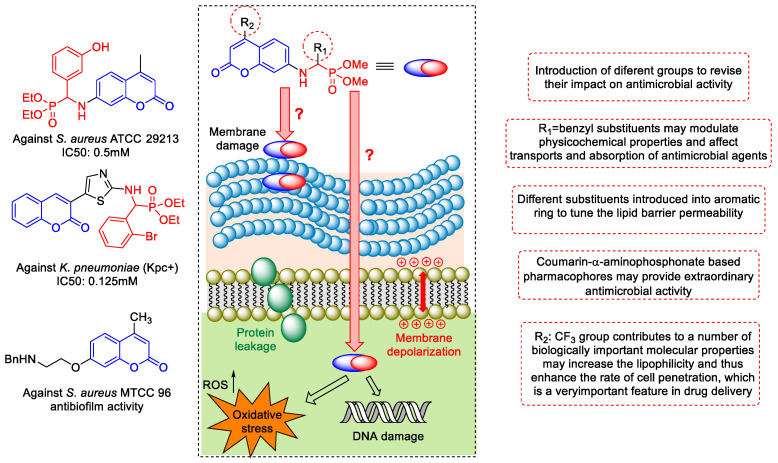
Biologically active antimicrobial derivatives of coumarin and α-aminophosphonate. Model of antimicrobial action on Gram-positive bacteria possessing the peptidoglycan (PG) layer.

**Figure 2 ijms-24-14886-f002:**
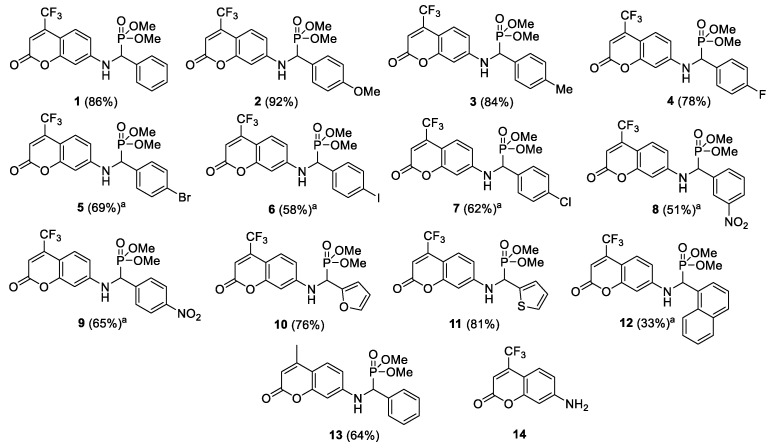
Structures of tested coumarin based α-aminophoshonates **1**–**13** and control 7-amino-4-trifluoromethylcoumarin **14**. ^a^ Reaction conducted in 2-Me-THF. Yields in brackets are provided for isolated products **1**–**13**.

**Figure 3 ijms-24-14886-f003:**
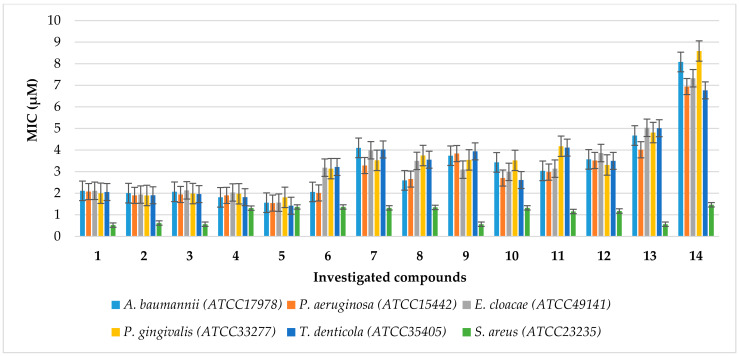
Minimum inhibitory concentration (MIC) of the investigated agents **1**–**14** in selected bacterial strains. The *x*-axis features compounds **1**–**14**. The *y*-axis shows the MIC value in µM. The order in which the compounds were applied to the plate is shown in Appendix A.

**Figure 4 ijms-24-14886-f004:**
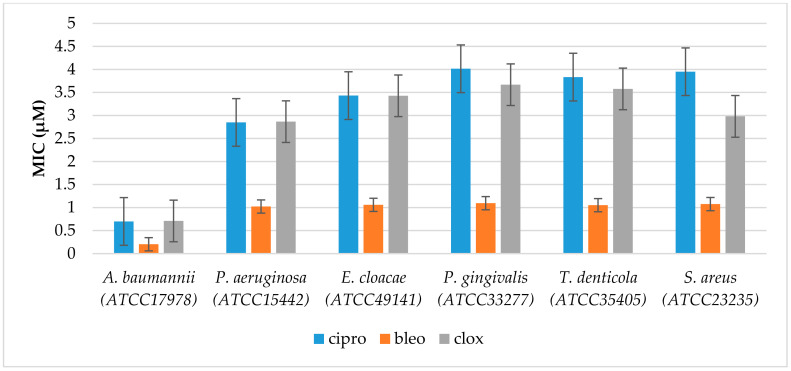
Examples of MIC in selected bacterial strains for studied antibiotics ciprofloxacin (cipro), bleomycin (bleo), and cloxacillin (clox). The *x*-axis features antibiotics used sequentially. The *y*-axis shows the MIC value in µM.

**Figure 5 ijms-24-14886-f005:**
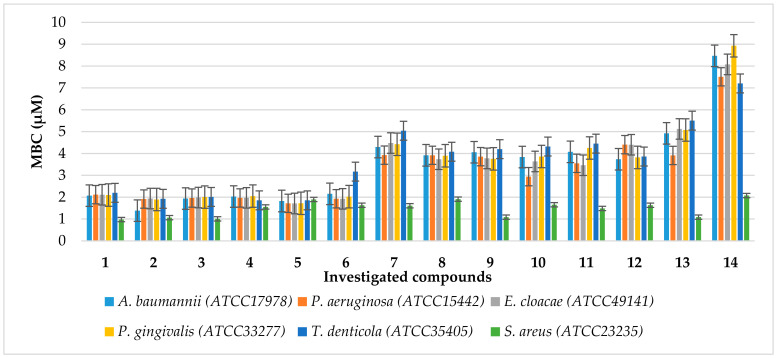
Minimum bactericidal concentration (MBC) of the investigated agents **1**–**14** in selected bacterial strains. The *x*-axis features compounds **1**–**14**. The *y*-axis shows the MBC value in µM. The order in which the compounds were applied to the plate is shown in Appendix A.

**Figure 6 ijms-24-14886-f006:**
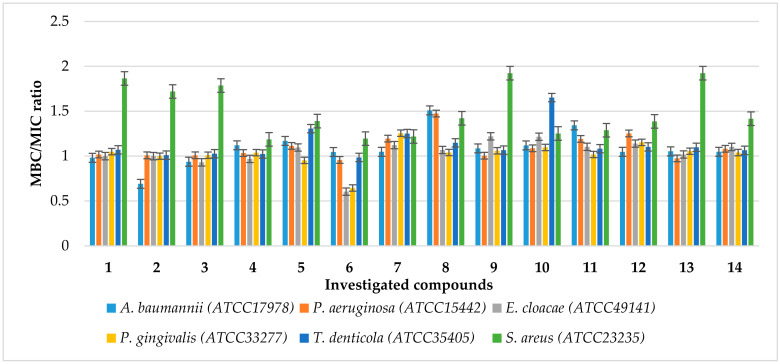
The ratio of MBC/MIC of the investigated agents **1**–**14** in selected bacterial strains. The *x*-axis features compounds **1**–**14**. The *y*-axis shows the MBC/MIC value in µM.

**Figure 7 ijms-24-14886-f007:**
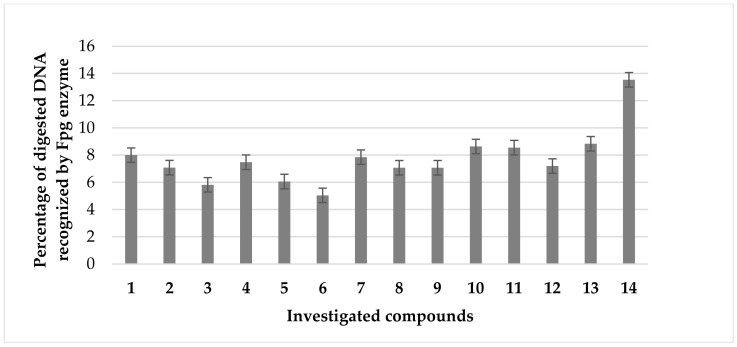
Percentage of plasmid DNA recognised by the Fpg enzyme (*y*-axis) with the bacterial strain on the sample of the strain *S. aureus* (ATCC 23235) (*x*-axis).

**Figure 8 ijms-24-14886-f008:**
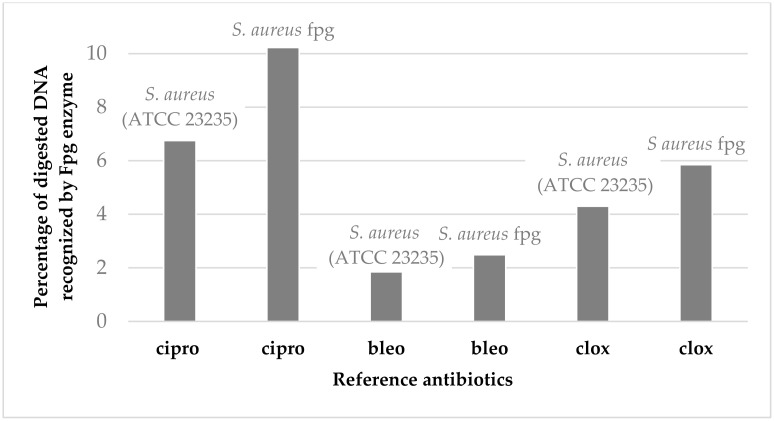
Percentage of bacterial DNA recognised by Fpg enzyme in an example of model bacterial strains of *S. aureus* (ATCC 23235) after treatment with ciprofloxacin, bleomycin, and cloxacillin.

**Table 1 ijms-24-14886-t001:** Statistical analysis of all compounds analysed by MIC, MBC, and MBC/MIC; <0.05 *, <0.01 **, <0.001 ***.

No. of Samples	4, 5, 6	7, 8, 10	11, 12, 14	Type of Test
*S. aureus* (ATCC 23235)	**	**	**	MIC
*S. aureus* (ATCC 23235)	**	**	***	MBC
*S. aureus* (ATCC 23235)	***	***	*	MBC/MIC

## Data Availability

On request of those interested.

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
