# Peer review of "Evaluation of Antibacterial Activity against Nosocomial Pathogens of an Enzymatically Derived α-Aminophosphonates Possessing Coumarin Scaffold"

_ijms, 2023, doi:10.3390/ijms241914886_

Round 1
Reviewer 1 Report
As the present manuscript currently exists, I would not recommend it for publication in a IJMC. Firstly, this material does not correspond to the level of the journal and almost completely repeats the research conducted earlier by the authors and published in this journal (reference 42 in the list of references). In addition, the text of the article, figures and supporting material contain a huge number of errors, typos, and inconsistencies, which makes reading the material very problematic. In the chemical part there is no detailed description, as it turns out, of already known and previously published compounds, but simply a figure from a previous article. The conclusions do not correspond to the content, just as it is unclear on what facts the authors’ conclusions about the mechanism of action of the compounds were made due to the abundance of graphs and figures with incomprehensible captions and the lack of a normal description in the discussion. More detailed descriptions of the research conducted and a more thorough approach by the authors to the design of the article and presentation of the material are needed.
There are a large number of typos. Some sentences repeat each other and very difficult to understand
Author Response
Dear Reviewer 1
Please find enclosed for your consideration a revised article entitled: “Evaluation of Antibacterial Activity Against Nosocomial Pathogens of an Enzymatically Derived α-Aminophosphonates Possessing Coumarin Scaffold." by Kowalczyk et al.
Firstly, we would like to express our gratitude to Reviewer 1 for all suggestions that allowed us to considerably improve our manuscript. To respond to the editor and reviewer queries we introduced a number of changes in the text. We have revised the text according to the suggestions and hope that you will now find it suitable for publication in IJMS journal.
Below, please find the detailed information on the changes in the manuscript with answers to all comments.
Rewiever 1: As the present manuscript currently exists, I would not recommend it for publication in a IJMC. Firstly, this material does not correspond to the level of the journal and almost completely repeats the research conducted earlier by the authors and published in this journal (reference 42 in the list of references). In addition, the text of the article, figures and supporting material contain a huge number of errors, typos, and inconsistencies, which makes reading the material very problematic. In the chemical part there is no detailed description, as it turns out, of already known and previously published compounds, but simply a figure from a previous article. The conclusions do not correspond to the content, just as it is unclear on what facts the authors’ conclusions about the mechanism of action of the compounds were made due to the abundance of graphs and figures with incomprehensible captions and the lack of a normal description in the discussion. More detailed descriptions of the research conducted and a more thorough approach by the authors to the design of the article and presentation of the material are needed.
Response: We are very grateful to the reviewer for his efforts in reviewing our manuscript. We have made every effort to eliminate editorial errors, lyricism and the like. The manuscript has been checked and corrected. In order to improve the quality of the manuscript, numerous paragraphs have been revised and rephrased. The manuscript focuses on the search for compounds with broad antibacterial activity, with particular emphasis on strains causing nosocomial infections. The research is based on the use of compounds that have shown promising activity against pathogenic E. coli strains. These compounds were prepared using biocatalysis, which allows obtaining substances free of metallic impurities, which is of particular importance in biological research and is also a requirement for the pharmaceutical industry. The conducted research has shown that the presence of characteristic functional groups is of great importance for the desired biological/antimicrobial activity. Comparative studies using known antibiotics have shown that the tested compounds are a promising alternative to known antibiotics, which is of great importance in the aspect of progressive multidrug resistance.

Reviewer 2 Report
REVIEWER'S REPORT
Manucsript title: Evaluation of Antibacterial Activity against Nosocomial Pathogens of an enzymatically derived α-Aminophosphonates possessing Coumarin scaffold (Authors: Paweł Kowalczyk, Dominik Koszelewski, Anna Brodzka, Karol Kramkowski and Ryszard Ostaszewski).
With a focus on gram-positive Staphylococcus aureus ATCC 23235 and gram-negative Enterobacter spp. that cause nosocomial infections, the goal of the study was to pinpoint the properties of the coumarin -amino phosphonate structure that might be significant for antibacterial activity against specific pathogenic strains.
This paper has the potential to be published in this journal, but only after significant and comprehensive corrections.
First and foremost, I advised that the whole text of the manuscript be corrected, condensed and paraphrased in many places, which would considerably improve the article's quality.
To my mind, Abstract is rather long and tedious. The same is true for Introduction. When reading the introduction, one gets the impression that this is review article rather than an experimental one. In the Introduction, I suggested simply cutting some parts of the text that include just basic information and are thereby overburdening the text. In my opinion, Figure 1 and the sentences (in lines 55-61) could be removed entirely. The statements in lines 37–55, as well as in lines 61–74, in lines 80–84, and in lines 116–120, should be rewritten or paraphrased. In overall, the Introduction should, in my opinion, be specifically linked to the content of this work in order to catch reader's attention.
In Results. Chemistry section. The first sentences (in lines 128-132) should be condensed and moved to Introduction. The sentence (in lines 139-141) have to be corrected/paraphrased. At the end of this section, it should be mentioned that NMR data are provided in Supplementary Materials.
In Cytotoxic studies. In this section, the basic broadly known information has to be removed by providing the major results achieved in this work, as well as their interpretation.
The whole text in Conclusion section should be shorten by highlighting only most important aspects defined in this work.
I'm not sure where in this work the combinatorial action of the compounds was examined, resulting in an improved (synergistic) effect, as described in Abstract (lines 13-15 ). The search for an enhanced effect is often based on combinatorial studies of two (or more) compounds, with the resulting data being examined by isobologram analysis or other methologies to prove the synergy.
Overall, this research is pretty significant in my opinion and should be published in this journal, but only after extensive restructuring, editing, and paraphrasing in multiple places. Squeezing "excess water" from the manuscript would make it far more appealing.
Although the entire text is comprehensible, substantial English editing, in my opinion, is needed
Author Response
Dear Reviewer 2
Please find enclosed for your consideration a revised article entitled: “Evaluation of Antibacterial Activity Against Nosocomial Pathogens of an Enzymatically Derived α-Aminophosphonates Possessing Coumarin Scaffold." by Kowalczyk et al.
Firstly, we would like to express our gratitude to Reviewer 2 for all suggestions that allowed us to considerably improve our manuscript. To respond to the editor and reviewer queries we introduced a number of changes in the text. We have revised the text according to the suggestions and hope that you will now find it suitable for publication in IJMS journal.
Below, please find the detailed information on the changes in the manuscript with answers to all comments.
Rewiever 2: This paper has the potential to be published in this journal, but only after significant and comprehensive corrections. First and foremost, I advised that the whole text of the manuscript be corrected, condensed and paraphrased in many places, which would considerably improve the article's quality.
Response: We are very grateful to the reviewer for his efforts in reviewing our manuscript. We have made every effort to eliminate editorial errors, typos. The manuscript has been checked and corrected. In order to improve the quality of the manuscript, numerous paragraphs have been revised and rephrased.
Rewiever 2: To my mind, Abstract is rather long and tedious. The same is true for Introduction. When reading the introduction, one gets the impression that this is review article rather than an experimental one. In the Introduction, I suggested simply cutting some parts of the text that include just basic information and are thereby overburdening the text. In my opinion, Figure 1 and the sentences (in lines 55-61) could be removed entirely. The statements in lines 37–55, as well as in lines 61–74, in lines 80–84, and in lines 116–120, should be rewritten or paraphrased. In overall, the Introduction should, in my opinion, be specifically linked to the content of this work in order to catch reader's attention.
Response: We are grateful for any comments from the Reviewer. In accordance with the suggestions, the Abstract and Introduction have been modified
Rewiever 2: In Results. Chemistry section. The first sentences (in lines 128-132) should be condensed and moved to Introduction. The sentence (in lines 139-141) have to be corrected/paraphrased. At the end of this section, it should be mentioned that NMR data are provided in Supplementary Materials.
Response: We are grateful for any comments from the Reviewer. The suggested changes have been incorporated into the manuscript
Rewiever 2: In Cytotoxic studies. In this section, the basic broadly known information has to be removed by providing the major results achieved in this work, as well as their interpretation.
Response: We are grateful for any comments from the Reviewer. The suggested changes have been made.
Rewiever 2: The whole text in Conclusion section should be shorten by highlighting only most important aspects defined in this work.
Response: We are grateful for any comments from the Reviewer. Concussions were revised and modified.
Rewiever 2: I'm not sure where in this work the combinatorial action of the compounds was examined, resulting in an improved (synergistic) effect, as described in Abstract (lines 13-15 ). The search for an enhanced effect is often based on combinatorial studies of two (or more) compounds, with the resulting data being examined by isobologram analysis or other methologies to prove the synergy.
Response: Thank you very much for this suggestion. We fully agree that the synergistic effect can bring very beneficial results. However, our goal was to determine the native activity of the tested compounds, we were also interested in which structural element of the tested agents is responsible for their activity. However, we will take the reviewer's attention into account when designing subsequent studies.
Rewiever 2: Overall, this research is pretty significant in my opinion and should be published in this journal, but only after extensive restructuring, editing, and paraphrasing in multiple places. Squeezing "excess water" from the manuscript would make it far more appealing.
Response: We are very grateful for the appreciation of our work and the positive potential of the research conducted

Reviewer 3 Report
Dear Authors,
I have several suggestions. First, I recommend you do a moderate English revision.
The manuscript specifies Figure S27 in Supplementary Materials, but it doesn’t exist. Likewise, the description of the x and y axes is irrelevant because the graphs are titles. I highly recommend performing the ANOVA and a posthoc test for all data results. Moreover, in the section “Materials and Methods” I suggest adding a subsection named “Statistical Analysis” and renaming the “Results” to “Results and Discussions”.
Another aspect that must be taken into account is the scientific name of the bacterial strains tested in this study. In this form of the manuscript, the authors have not written the names of the strains correctly and I propose using their abbreviations. Also, please pay attention to the "Gram-positive" and "Gram-negative" writing! There is no such thing as "gram-minus" or "gram-positive". The name of this classification is given after a Danish scientist named Hans Christian Gram. ;)
After reading and comprehending the information presented in the paper, I highly recommend it for publication after minor revisions.
I recommend you do a moderate English revision.
Author Response
Dear Reviewer 3
Please find enclosed for your consideration a revised article entitled: “Evaluation of Antibacterial Activity Against Nosocomial Pathogens of an Enzymatically Derived α-Aminophosphonates Possessing Coumarin Scaffold." by Kowalczyk et al.
Firstly, we would like to express our gratitude to Reviewer 3 for all suggestions that allowed us to considerably improve our manuscript. To respond to the editor and reviewer queries we introduced a number of changes in the text. We have revised the text according to the suggestions and hope that you will now find it suitable for publication in IJMS journal.
Below, please find the detailed information on the changes in the manuscript with answers to all comments.
Rewiever 3: I have several suggestions. First, I recommend you do a moderate English revision.
Response: We are very grateful to the reviewer for his efforts in reviewing our manuscript. We have made every effort to eliminate editorial errors, typos. The manuscript has been checked and corrected. In order to improve the quality of the manuscript, numerous paragraphs have been revised and rephrased.
Rewiever 3: The manuscript specifies Figure S27 in Supplementary Materials, but it doesn’t exist. Likewise, the description of the x and y axes is irrelevant because the graphs are titles. I highly recommend performing the ANOVA and a posthoc test for all data results. Moreover, in the section “Materials and Methods” I suggest adding a subsection named “Statistical Analysis” and renaming the “Results” to “Results and Discussions”.
Response: We are very sorry, the mentioned figure has been added to the supplement. Following the reviewer's recommendation, we have added a section on Statistical Analysis.
All experimental data were presented as the means ± standard error of the mean (SEM, manufacturer, Saint Louis, MO, USA) of a minimum of three independent experiments (n = 3). The Tukey test was used Statistical significance is indicated by (p < 0.05): * p < 0.05, ** p < 0.1, *** p < 0.01. The Tukey post-hoc test is one of the most commonly used tests for comparing pairs of means. It can be used for different sample sizes. It is based on a distribution called the "studentized range statistic" The Tukey method is more conservative than the NIR test, but less conservative than the Scheffé test. The experimental error rate for all pairwise comparisons remains at the set error level, which means that if an α0.05 level of statistical significance is assumed for the ANOVA test, the same level of statistical significance will be used for all comparisons between pairs (samples). This procedure is used in a situation where the assumption of equality of variances in the samples is met. If we wanted to compare two groups, we would use an independent samples t test; however, if we wanted to compare two variables regarding the same individuals (observations), we would use the t test for dependent samples. This distinction between dependent and independent groups is also important for the ANOVA method.
The section Results was remained to Results and Discussions
Rewiever 3: Another aspect that must be taken into account is the scientific name of the bacterial strains tested in this study. In this form of the manuscript, the authors have not written the names of the strains correctly and I propose using their abbreviations. Also, please pay attention to the "Gram-positive" and "Gram-negative" writing! There is no such thing as "gram-minus" or "gram-positive". The name of this classification is given after a Danish scientist named Hans Christian Gram. ;)
Response: Thank you very much for this valuable attention. We are very sorry for this mistake, we completely agree. The names of microorganisms have been corrected and appropriate abbreviations have been used
Rewiever 3: After reading and comprehending the information presented in the paper, I highly recommend it for publication after minor revisions.
Response: We are very grateful for the appreciation of our work and the positive potential of the research conducted

Round 2
Reviewer 1 Report
The authors have revised the manuscript, and although I still do not see the relevance of this study due to the lack of novelty in the structures of previously described compounds and their toxicity, the manuscript can be recommended for publication after minor changes (the text contains typos (for example, line 153), unnecessary spaces, etc.). In addition, I consider it necessary to indicate in the abstract that the compounds described in this work were obtained previously. It would also be great if there was a discussion of the influence of the structure of aminophosphonate compounds 1-13 among themselves, and not just in comparison with the CF3-derivative 14.
Author Response
Please find enclosed for your consideration a revised article ijms-2635160 entitled: “Evaluation of Antibacterial Activity Against Nosocomial Pathogens of an Enzymatically Derived α-Aminophosphonates Possessing Coumarin Scaffold." by Kowalczyk et al.
Firstly, we would like to express our gratitude to Reviewers for their suggestions that allowed us to considerably improve our manuscript. To respond to the editor and reviewers queries we introduced a number of changes in the text. We have revised the text according to the suggestions and hope that you will now find it suitable for publication in IJMS journal.
Below, please find the detailed information on the changes in the manuscript with answers to all comments.
Rewiever 1: The authors have revised the manuscript, and although I still do not see the relevance of this study due to the lack of novelty in the structures of previously described compounds and their toxicity, the manuscript can be recommended for publication after minor changes (the text contains typos (for example, line 153), unnecessary spaces, etc.). In addition, I consider it necessary to indicate in the abstract that the compounds described in this work were obtained previously. It would also be great if there was a discussion of the influence of the structure of aminophosphonate compounds 1-13 among themselves, and not just in comparison with the CF3-derivative 14.
Response: We are very grateful to the reviewer for his efforts in reviewing our manuscript. We have made every effort to eliminate typos. According to the reviewer's suggestion, a discussion was carried out on the influence of the structure of the tested compounds on their antimicrobial activity and selectivity

Reviewer 2 Report
When reading the text, it is clear that the authors made numerous revisions. However, some additional corrections are required.
Especially catches the eye the sub-section 3.3 (Statistical analysis, lines 286-300). In my opinion, it should be condensed into a few sentences, with reference(s), and might be written as follows: "All experimental data from at least three different trials (n = 3) were given as means standard error (SEM, manufacturer, Saint Louis, MO, USA).To compare pairs of means, the Tukey post-hoc test was used, indicating statistical significance with * p < 0.05, ** p < 0.1, and *** p < 0.01 [reference(s)????]." The remaining (redundant) text in this sub-section should be removed.
In my opinion, something is amiss with the semantics in the legends of Figures 7 and 8 (how can compounds be statistically reliable ?).
This manuscript could be accepted after minor revisions.
Minor revisions are required.
Author Response
Please find enclosed for your consideration a revised article ijms-2635160 entitled: “Evaluation of Antibacterial Activity Against Nosocomial Pathogens of an Enzymatically Derived α-Aminophosphonates Possessing Coumarin Scaffold." by Kowalczyk et al.
Firstly, we would like to express our gratitude to Reviewers for their suggestions that allowed us to considerably improve our manuscript. To respond to the editor and reviewers queries we introduced a number of changes in the text. We have revised the text according to the suggestions and hope that you will now find it suitable for publication in IJMS journal.
Below, please find the detailed information on the changes in the manuscript with answers to all comments.
Rewiever 2: When reading the text, it is clear that the authors made numerous revisions. However, some additional corrections are required.
Especially catches the eye the sub-section 3.3 (Statistical analysis, lines 286-300). In my opinion, it should be condensed into a few sentences, with reference(s), and might be written as follows: "All experimental data from at least three different trials (n = 3) were given as means standard error (SEM, manufacturer, Saint Louis, MO, USA).To compare pairs of means, the Tukey post-hoc test was used, indicating statistical significance with * p < 0.05, ** p < 0.1, and *** p < 0.01 [reference(s)????]." The remaining (redundant) text in this sub-section should be removed. In my opinion, something is amiss with the semantics in the legends of Figures 7 and 8 (how can compounds be statistically reliable ?).
Response: We are very grateful to the reviewer for his efforts in reviewing our manuscript. As suggested, the section on statistical analysis has been modified and the descriptions of the Figures have been corrected.
